# ZGSO Spinel Nanoparticles with Dual Emission of NIR Persistent Luminescence for Anti-Counterfeiting Applications

**DOI:** 10.3390/ma16031132

**Published:** 2023-01-28

**Authors:** Guanyu Cai, Teresa Delgado, Cyrille Richard, Bruno Viana

**Affiliations:** 1Université PSL, Chimie ParisTech, CNRS, IRCP, Institut de Recherche de Chimie Paris, 75005 Paris, France; 2Université Paris Cité, CNRS, INSERM, UTCBS, Unité de Technologies Chimiques et Biologiques pour la Santé, Faculté de Pharmacie, 75006 Paris, France

**Keywords:** spinel, photoluminescence, persistent luminescence, nanoparticles, anti-counterfeiting, NIR

## Abstract

The property of persistent luminescence shows great potential for anti-counterfeiting technology and imaging by taking advantage of a background-free signal. Current anti-counterfeiting technologies face the challenge of low security and the inconvenience of being limited to visible light emission, as emitters in the NIR optical windows are required for such applications. Here, we report the preparation of a series of Zn_1+x_Ga_2−2x_Sn_x_O_4_ nanoparticles (ZGSO NPs) with persistent luminescence in the first and second near-infrared window to overcome these challenges. ZGSO NPs, doped with transition-metal (Cr^3+^ and/or Ni^2+^) and in some cases co-doped with rare-earth (Er^3+^) ions, were successfully prepared using an improved solid-state method with a subsequent milling process to reach sub-200 nm size particles. X-ray diffraction and absorption spectroscopy were used for the analysis of the structure and local crystal field around the dopant ions at different Sn^4+^/Ga^3+^ ratios. The size of the NPs was ~150 nm, measured by DLS. Doped ZGSO NPs exhibited intense photoluminescence in the range from red, NIR-I to NIR-II, and even NIR-III, under UV radiation, and showed persistent luminescence at 700 nm (NIR-I) and 1300 nm (NIR-II) after excitation removal. Hence, these NPs were evaluated for multi-level anti-counterfeiting technology.

## 1. Introduction

Photoluminescence has become an increasingly important imaging technology, with popular applications in many fields including lasers, sensors, lighting, and bio-probe imaging [1,2,3]. Anti-counterfeiting also employs photoluminescent materials due to their high throughput, low cost, flexibility, and stability [4,5]. Current optical-based anti-counterfeiting technologies are mainly based on down-conversion or up-conversion luminescence under excitation [5,6,7,8,9,10,11,12]. However, real-time fluorescence of lanthanides for anti-counterfeiting purposes is susceptible to producing interference with the background and excitation light [13]. In this context, the development of a background-free anti-counterfeiting technology is highly desirable [14,15]. To avoid background noise, near-infrared (NIR) or short-wavelength infrared (SWIR) luminescence imaging has begun to be developed as a result of InGaAs-based NIR detectors that can be employed not only for anti-counterfeiting applications but also to provide images with better clarity and spatial resolution [16,17,18,19,20].

The optical window in NIR (or SWIR) ranges can be artificially divided into deep-red NIR-I (700–950 nm), for which a Si-based detector could be used, and NIR-II (1000–1400 nm) and NIR-III (1500–1700 nm) subregions, for which an InGaAs-based detector is required [21,22,23]. However, current anti-counterfeiting technologies face the challenge of low security and the inconvenience of being limited to visible light emission as emitters in the second and even third optical windows are required [24,25]. Ni^2+^ and Er^3+^ ions, for example, could play an excellent role as emitters due to their luminescence in NIR-II and NIR-III [24,25,26,27,28]. Indeed, previous studies in the field have proposed the use of RE^3+^ ions in Sn-free ZGO NPs as sensitizers to enhance the NIR-I persistent luminescence of Cr^3+^ at 700 nm [29,30,31,32,33]. We have reported NIR-I persistent luminescence of Cr^3+^ ions in bulk ZGSO ceramics corresponding to the ^2^E → ^4^A_2_ (^4^F) transition [34], and Qiu et al. were able to tune the optical properties of ZGSO by precisely controlling the crystal field by use of Sn in these bulk materials [35].

In this study, we report the synthesis of a series of nanomaterials based on an Ni^2+^-doped Zn_1+x_Ga_2−2x_Sn_x_O_4_ (ZGSO) matrix with varying amounts of Sn (from x = 0 to x = 0.6). After selecting the most favorable composition in terms of duration and intensity of persistent luminescence, the effect of co-dopant Cr^3+^ and/or rare-earth (Er^3+^) ions was evaluated, in order to provide tunable multi-emission from deep red to NIR-III. At the nanoscale, these materials (ZGSO) can be highly stable in an aqueous medium for several days, which allows their use in anti-counterfeiting, as the patterns can be easily machine printed or hand painted with the aid of an aqueous solvent. By introducing a “multi-level” anti-counterfeiting method based on ZGSO persistent luminescence NPs with emission in multiple ranges, we could effectively cover the deep-red to NIR-III ranges. New anti-counterfeiting systems with persistent luminescence nanoparticles could be developed with various duration times. The detection tools being employed are Si-based cameras/detectors for detection in deep-red to NIR-I ranges and InGaAs-based cameras/detectors for NIR-II and III ranges. Indeed, ZGSO NPs not only expand the area of anti-counterfeiting technology but also provide higher security compared to traditional phosphors emitting visible light and detectable with the naked eye.

## 2. Materials and Methods

### 2.1. Materials

Materials were ZnO (99.99%, Strem Chemicals,), Ga_2_O_3_ (99.99%, Strem Chemicals), SnO_2_ (99.99%, Strem Chemicals), and dopant element sources were NiO (99.99% Aldrich Chemistry, St. Louis, MO, USA), Er_2_O_3_ (99%, Alfa Aesar, Haverhill, MA, USA), and Cr_2_O_3_ (99.99%, Aldrich Chemistry). All chemicals were used as received without further purification.

### 2.2. Synthesis of Persistent Luminescence Phosphors

The powder samples were synthesized by a high-temperature solid-state method starting from binary oxides. The raw material mixture was prepared according to the stoichiometric ratio of the compounds. For the example of ZGSO-3:Ni^2+^ (Zn_1.3_Ga_1.39_Ni_0.01_Sn_0.3_O_3.995_), 1.3 mmol of ZnO, 1.3 mmol of Ga_2_O_3_, 0.3 mmol of SnO_2_, and 0.1 mmol of NiO were weighed and ground homogenously in an agate mortar, and the mixtures were introduced into alumina crucibles. The alumina crucible was subsequently placed in a high-temperature furnace at 1300 °C for 6 h in the air to produce the final sample [36]. After cooling to room temperature, the phosphors were ground to fine powders. The other samples in this paper were prepared using the same method.

### 2.3. Recovery of Persistent Luminescence NPs

The ZGSO powders (~500 mg) were crushed using a Pulverisette 7 Fritsch Planetary Ball Mills, with dropped 1 mL of 5 mM HCl solution, at a speed of 1000 rpm for 2–4 h, to reduce their size. The different materials were then transferred into a flask and stirred vigorously for 24 h at room temperature. The final ZGSO NPs were obtained from polydisperse colloidal mixtures by centrifugation with a SANYO MSE Mistral 1000 at 3500 rpm for 5 min. ZGSO NPs were recovered from the supernatant with a size of ~150 nm measured by Dynamic Light Scattering (DLS), as shown in Appendix A. The supernatants were gathered and concentrated to a final ~5 mg/mL suspension.

### 2.4. Characterization

#### 2.4.1. X-ray Diffraction (XRD)

X-ray diffraction (XRD) patterns of the materials showed crystalline phases with cubic spinel structures. XRD was performed with an X-ray diffractometer (XPert PRO, PANalytical, Malvern Panalytical Ltd., Malvern, UK) equipped with a Ge111 single-crystal monochromator and by selecting the K_α1_ radiation wavelength of the Cu X-ray tube (0.15405 nm).

#### 2.4.2. Dynamic Light Scattering (DLS)

The hydrodynamic diameter of the ZGSO NPs was characterized by dynamic light scattering (DLS) performed with a Zetasizer Nano ZS (Malvern Instruments, Southborough, MA, USA) equipped with a 632.8 nm helium–neon laser and 5 mW power, with a detection angle of 173° (non-invasive backscattering).

#### 2.4.3. Absorption Spectroscopy

Absorption measurements of the dry ZGSO NPs were carried out in a UV/Vis/NIR spectrophotometer (Varian Cary 6000 i, Agilent). The resolution was 0.1 nm for the Cr bands.

#### 2.4.4. Photoluminescence

NIR photoluminescence (PL) measurements of dry ZGSO NPs were performed with an NIR camera (PyLoNIR, Princeton Instruments, Trenton, NJ, USA) for the NIR-II range cooled at −100 °C and coupled to a monochromator (Acton Spectra Pro, Princeton Instruments), with 300 grooves per mm and centered at 1200 nm.

Visible and deep-red (or NIR-I) photoluminescence (PL) measurements of dry ZGSO NPs were performed using a CCD camera (Roper Pixis 100, Princeton Instruments) cooled at −65 °C and coupled to a monochromator, with 300 grooves per mm and centered at 500 nm.

#### 2.4.5. Persistent Luminescence

The dry ZGSO NPs samples were thermally detrapped before each experiment and then kept in the dark. The samples were loaded with a 365 nm lamp for 5 min at 290 K, and after excitation removal, the afterglow was recorded for 15 min at the same temperature. The signal was followed with the same camera as in the PL experiment. Afterglow curves were obtained by integrating the intensity of the persistent luminescence spectra as a function of time.

#### 2.4.6. Anti-Counterfeiting Applications

Anti-counterfeiting patterns were hand painted using a ZGSO NP suspension (10 mg of NPs into 1 mL of water), dropped onto paper to form anti-counterfeiting marks, and then dried at room temperature (RT) for 24 h. Using this method, the “UTCBS” acronym was marked using ZGSO-3:Ni^2+^, Er^3+^ NPs. The other “MPOE” acronym was marked using ZGSO-3:Ni^2+^, Er^3+^, Cr^3+^ NPs. Thus, the tests became fluorescent signals able to provide an effective property of persistent luminescence. These could be employed in anti-counterfeiting applications by creating persistent luminescence images.

Imaging in the NIR-I range was recorded with a photon-counting device (Optima, Biospace Lab, Nesles-la-Vallée, France). The persistent luminescence images of the various figures were recorded for 5 min after removal of the 365 nm lamp excitation.

Imaging signals in the NIR-II range were recorded with an InGaAs camera (Princeton NIRvana camera). The persistent luminescence features were recorded after removal of the 365 nm lamp excitation. The camera collected an image every 3 min (during the 15 min period following excitation removal). In addition, the PL images of the patterns were recorded with the camera under various excitations ranging from 365 nm (lamp) to 808 nm and 980 nm (NIR laser).

## 3. Results and Discussion

A normal spinel (ZGO) and an inverse spinel (ZSO), in addition to a series of complex spinel nanomaterials, were designed as combinations of the normal spinel ZnGa_2_O_4_ and inverse spinel Zn_2_SnO_4_ with different Sn amounts, as represented by the formula Zn_1+x_Ga_2−2x_Sn_x_O_4_, x = 0, 0.1, 0.2, 0.3, …, 0.6, denoted as ZGSO (from ZGSO-0 to ZGSO-6, respectively). The content of Ni^2+^-doped ZGSO NPs can be seen in Appendix A. The most favorable persistent luminescence properties suggested a nominal doping content for the activator Ni^2+^ at 0.5 mol % in respect to Ga^3+^ ions in the matrix; this will be discussed in detail in the following sections of the paper. Divalent nickel ions as dopants in this host are in the octahedral site, as the stabilization energy is fairly important (greater than 80 kJ.mol^−1^) [37]. Therefore, only the octahedral sites are populated with Ni^2+^ ions as dopants. Furthermore, one should notice the very similar ionic radius of Ni^2+^ with Ga^3+^ cations in addition to the strong octahedral coordination preference [38]. In addition, other contents of doping cations (Ni^2+^, Er^3+^, and Cr^3+^) in ZGSO-3 NPs are shown in Appendix A. Thermoluminescence spectra of Ni^2+^ in ZGO and ZGSO matrices have also been obtained; these are shown in Appendix A. As previously observed for ZGSO:Cr [34], the TL glow presents shallower traps when Sn^4+^ is introduced with an increase in signal (see the signal-over-noise ratio) and the resulting broadening.

### 3.1. Crystal Structure Analysis by Powder X-ray Diffraction (PXRD)

The ionic radius of Sn^4+^ is 0.83 Å, which is slightly greater than that of Ga^3+^ (0.76 Å). Thus, the Sn^4+^ ion is well adapted to an octahedral configuration with Sn–O distances (2.05 Å), which is similar to the Ga–O distances (1.98 Å). Sn^4+^ ions can be introduced into the ZnGa_2_O_4_ host [34] without affecting its structural properties. As shown in Figure 1a, the diffraction peaks (at 2*θ* = 18.4°, 30.3°, 35.7°, 57.4°, 63.0°) of sample ZGSO-0 (Sn free, normal ZGO) relate well to the standard lattice planes [(111), (220), (311), (511), (440), respectively] of ZnGa_2_O_4_ (JCPDS No. 01-086-0413). For the series of Zn_1+x_Ga_2−2x_Sn_x_O_4_ materials with increasing x values (concentration of Sn), the PXRD patterns show a slight shift in the diffraction peaks, as seen in Figure 1. The diffraction peaks shift to lower 2*θ* values with an increase in Sn^4+^/Ga^3+^ ratio. A comparison of the ZGSO-2 (x = 2) and ZGSO-4 (x = 4) patterns is shown in Appendix A.

The Ni^2+^ cation crystal field could be modified by tuning the Sn^4+^/Ga^3+^ ratios, as seen in the following section of the paper. This is also the case for the introduction of Cr^3+^, as previously reported [35]. In contrast, any shift in the position of the diffraction peaks is observed with a 0.5% Ni^2+^ doping ratio (Figure 1b). Ni^2+^, Cr^3+^, and to a lesser extent Er^3+^, as the coordination is less favorable, are cations that can be incorporated effectively (up to 1 or 2%, as shown previously) into octahedral lattice sites, due to a similar ionic radius and valence with Ga^3+^ cations [39]. Hence, as it has been suggested that multi-emission is effectively covered in the range from visible to NIR-III, low doping ratios of Er^3+^ (1%) and Ni^2+^ (0.5%) or Cr^3+^ (0.5%) have been introduced into hosts without concern for the stability of the crystal structure and while retaining the normal spinel structure of ZnGa_2_O_4_ in all samples regardless of the tested Sn^4+^ contents, as seen in Figure 1.

### 3.2. Absorption Spectra Analysis

The absorption spectra of the samples were recorded to further validate the relationship between the change in crystal structure and the energy difference between electronic transitions. ZGSO-1, ZGSO-3, and ZGSO-5 are shown in Figure 2a. Appendix A shows the normalized absorption spectra of Ni^2+^-doped ZGSO-2 and ZGSO-4. Strong, broad excitation bands located between ~260 nm and ~320 nm are observed. For example, in the ZGSO-3: Ni sample, the observed shoulders at around 370 nm are attributed to the ^3^A_2_ (^3^F) → ^3^T_1_ (^3^P) spin-allowed transition of Ni^2+^, almost overlapping with the bandgap edge of the ZGSO matrix. This bandgap absorption edge originates from the O–Ga charge-transfer transition in the ZGSO host, which is also seen in Figure 2b, and the bandgap values decrease with Sn^4+^ content. The bandgap energy values can be extracted and calculated using the Kubelka–Munk function [35]. With an increase in Sn^4+^ content in the ZGSO matrix, the bandgap values varied from 3.5 to 4.6 eV using the following formula:$$E=\frac{hc}{\lambda }$$
where *E* is the bandgap energy (in eV) and *λ* is the experimental absorption wavelength, as shown in Figure 2.

In addition, the d^8^ Tanabe–Sugano diagrams shown in Figure 2c and Appendix A effectively show that the shifted absorption peaks of Ni^2+^ towards a longer wavelength depend on an increase in the Sn^4+^/Ga^3+^ ratio corresponding to a decrease in crystal field strength [40]. Two absorption peaks of ZGSO-3: Ni^2+^ observed at ~600 nm and ~1050 nm originate from the ^3^A_2_ (^3^F) → ^3^T_1_ (^3^F) and ^3^A_2_ (^3^F) → ^3^T_2_ (^3^F) transitions of Ni^2+^, respectively. For a more quantitative analysis, the crystal field strength value *D_q_* and the Racah parameter *B* can be estimated using the energies of various absorption peaks as follows [29]:(1)$$10\;D_{q}=v_{2\;}$$
(2)$$\frac{v_{1}^{2}+2\;v_{3}^{2}-3\;v_{1\;}v_{3\;}}{15\;v_{1}-27\;v_{3}}$$
where $v_{1}$ and $v_{3}$ are the energies corresponding to Ni^2+^ ions [^3^A_2_ (^3^F) → ^3^T_1_(^3^P)] and [^3^A_2_ (^3^F) → ^3^T_2_ (^3^F)] transitions, respectively.

As a result, variation in the crystal field strength of ZGSO was observed. As Sn^4+^ content varied from 0.1 to 0.3, and even up to 0.5, the energy level of the divalent nickel corresponded to lower crystal field strength values (10 *D_q_* decreased from 10.58 to 10.12). Crystal field values, $v_{3}\;$(10 *D_q_*/*B*), in addition to other related parameters, are shown in Table 1.

Furthermore, Er^3+^ and Cr^3+^ ions were used as dopants to obtain other emission wavelengths. The absorption spectra of Er^3+^ and Er^3+^/Cr^3+^ co-doped ZGSO: Ni^2+^ NPs are shown in Figure 2d. The absorption peak at 1530 nm is attributed to the Er^3+ 4^I_15/2_ → ^4^I_13/2_ transition. The broad absorption bands at 410 nm and 570 nm are attributed to the ^4^A_2_ (^4^F) →^4^T_1_ (^4^F) and ^4^A_2_ (^4^F) → ^4^T_2_ (^4^F) transitions of Cr^3+^, respectively.

### 3.3. Photoluminescence (PL) Spectral Analysis

Using a control of tin composition, a tunable ^3^T_2_ (^3^F) → ^3^A_2_ (^3^F) transition of Ni^2+^ PL extends from 1270 to 1340 nm, as shown in Figure 3a. The variation in wavelength is also effectively explained in the Tanabe–Sugano diagram d^8^ in Figure 2d.

As co-dopants, Er^3+^ cations with several emissions in the visible, NIR-I, and even SWIR range may be very interesting, and we have focused our attention on the ZGSO-3: Ni^2+^, Er^3+^ sample, which presents the most favorable persistent luminescence properties, as shown in the following section. As shown in Figure 3b, ZGSO-3: Ni^2+^, Er^3+^ NPs show PL peaks at 980 nm (NIR-I range) and 1300 nm and 1533 nm (NIR-II range) under 365 nm irradiation. The peak at 1300 nm is attributed to the ^3^T_2_ (^3^F) → ^3^A_2_ (^3^F) transition of octahedral Ni^2+^ ions [41].

On the other hand, Er^3+^ is responsible for the emission at 980 nm (NIR-I) and 1533 nm (NIR-II), attributed to the ^4^I_11/2_ → ^4^I_15/2_ and ^4^I_13/2_ → ^4^I_15/2_ transitions, respectively, as shown in Appendix A. These optical properties—with emission in the visible, deep-red, and NIR ranges—have been obtained due to knowledge within our laboratories, which have been working on the zinc gallate matrix for several years [42,43,44,45,46,47]. The PL spectra of (a) ZGSO:Cr^3+^, (b) ZGSO:Er^3+^, and (c) ZGSO:Cr^3+^, Er^3+^ in the visible range are shown in Figure 4. ZGSO-3: Ni^2+^, Er^3+^, Cr^3+^ NPs were also considered within this work because Cr^3+^ provides a 700 nm persistent luminescence in the deep-red (or NIR-I) range [46,47]. Under 365 nm UV excitation, the multi-emission on the PL spectrum of ZGSO-3: Ni^2+^, Er^3+^, Cr^3+^ NPs covers visible, NIR-I, NIR-II, and NIR-III ranges; the PL features will be compared to the persistent luminescence properties in the following section of the paper.

### 3.4. Persistent Luminescence Spectral Analysis

After UV excitation removal, a persistent luminescence signal is observed for all Ni^2+^-doped ZGSO NPs. This persistent luminescence is assigned to the ^3^T_2_ (^3^F) → ^3^A_2_ (^3^F) transition of octahedral Ni^2+^ cations [42]. The position of the broad emission peak of Ni^2+^ shifts from 1270 to 1340 nm by increasing the Sn (x value, Appendix A) content in the matrix (see Figure 5a), due to a decrease in the strength of the crystal field. Appendix A shows a possible mechanism to explain the persistent luminescence signal for ZGSO-3: Ni^2+^ NPs. Under UV excitation, charges are formed and trapped in the ZGSO host defects. Based on this system, the release with temperature of the trapped charges after excitation removal and recombination toward the Ni^2+^ centers lead to the persistent luminescence signal in the SWIR range, attributed to the ^3^T_2_ (^3^F) → ^3^A_2_ (^3^F) transition (Appendix A). An increase in Zn^2+^ and Sn^4+^ content leads to a decrease in crystal field around Ni^2+^, similar to that seen with photoluminescence, and a shift in the persistent luminescence signal towards a longer wavelength (i.e., lower energy) is observed, as seen in Figure 5a.

Although all samples show a rapid decrease in persistent luminescence, the initial intensity depends on the sample. For example, as shown in Figure 5b, of ZGSO-1:Ni^2+^, ZGSO-3:Ni^2+^, and ZGSO-5:Ni^2+^, ZGSO-3:Ni^2+^ (namely, Zn_1.3_Ga_1.4_Sn_0.3_O_4:_:0.5%Ni^2+^) shows the highest initial intensity value of persistent luminescence at 1300 nm, as shown in Appendix A. Furthermore, the Ni^2+^ doping concentration was optimized (Appendix A) and was revealed to be 0.5%; in comparison, the persistent luminescence intensity of the Ni^2+^-doped ZGSO-3 samples with different nickel concentrations was 0.25%, 0.5%, and 1%, as shown in Appendix A. As a result, the persistent luminescence at 1300 nm increased as the crystal field value (10 *D_q_*/*B)* decreased to 10.20 from an initial value of 10.58. Zn_1.3_Ga_1.4_Sn_0.3_O_4:_:0.5%Ni^2+^ is the optimal material as it has the strongest persistent luminescence intensity of all the samples prepared and reported in Appendix A.

The effect of the amount of Er^3+^ in the ZGSO matrix was also evaluated. As shown in Appendix A, of the different concentrations tested, the optimal amount of Er^3+^ is 1% atomic. On the other hand, the inset in Figure 5b shows the time dependence of the NIR persistent luminescence with wavelengths at 1300 nm and 1530 nm from Er^3+^, Ni^2+^ co-doped ZGSO-3 NPs. While the persistent luminescence intensity at 1300 nm reduced slowly, the emission at 1530 nm (see Figure 3b) fully disappeared when the excitation was removed, and no persistent emission could be recorded. For the ZGSO:Cr^3+^, Er^3+^ samples, the visible persistent emissions in the red and deep-red ranges are shown in Appendix A. Persistent luminescence spectra of ZGSO:Cr^3+^ and ZGSO:Cr^3+^, Er^3+^ are similar, and no red emission is observed (at 650 nm for Er^3+^ cations); the persistent luminescence under UV excitation is governed solely by Cr^3+^ decay. Meanwhile, Appendix A shows schematic diagrams of the absorption/emission bands; trapping/detrapping and energy-transfer mechanisms lead to deep-red emissions of (a) ZGSO-3:Cr^3+^ NPs, NIR emission of (b) ZGSO-3:Ni^2+^ NPs, and 4f-4f transition of (c) ZGSO-3:Er^3+^ NPs.

The spectra in Figure 6 indicate the PL (Figure 6a) and persistent luminescence (Figure 6b) of ZGSO:Ni^3+^, Cr^3+^, Er^3+^, detected with an Si-based detector. In the SWIR (from NIR-II to NIR-III) range; Figure 6c,d also provide the spectrum of PL (Figure 6c) and persistent luminescence (Figure 6c) of the same sample ZGSO:Ni^2+^, Cr^3+^, Er^3+^, detected with an InGaAs-based detector. Peaks of Er^3+^ vanished at ~550 nm and ~660 nm in the visible range and at ~980 nm and ~1530 nm in the NIR, corresponding to a lack of Er^3+^ remaining in the samples. With these features of PL and persistent luminescence, anti-counterfeiting applications have been proposed.

### 3.5. Towards Anti-Counterfeiting Applications

#### 3.5.1. A Two-Step Anti-Counterfeiting Method in the NIR-II Range

A new two-step anti-counterfeiting method in the NIR range based on ZGSO:Ni^2+^, Er^3+^ NPs is introduced. As described above, NIR multi-emissions of PL at ~980 nm, ~1300 nm, and ~1530 nm are obtained from NPs under UV irradiation as a first security step (for bright emissions, UV excitation wavelengths should be in the range 275–400 nm). As shown by the example image, the “UTCBS” acronym noted by ZGSO:Ni^2+^, Er^3+^ NPs can be easily detected due to their strong PL (Figure 7a), corresponding to the PL in the NIR range.

For a second security step, after removing UV excitation, the persistent luminescence pattern “UTCBS” is obtained with an InGaAs-based camera for at least 15 min (Figure 7b). Additionally, another security step could be observation of the change in signal shape between PL and persistent luminescence (see Figure 3b). The Er^3+^ emission in the SWIR vanishes in the persistent luminescence spectrum, which provides a new step in the field of anti-counterfeiting, NIR (or SWIR) emission giving higher accuracy and security. In addition, the PL of the pattern “UTCBS” noted by ZGSO:Ni^2+^, Er^3+^ NPs excited by the 980 nm and 808 nm NIR lasers, respectively, in the Er^3+^ absorption bands is shown in Figure 7. No persistent luminescence was observed under these NIR excitation wavelengths.

#### 3.5.2. Multi-Level Anti-Counterfeiting Method in Visible, NIR-I, and NIR-II Ranges

To advance this proof of concept using Zn_1.3_Ga_1.4_Sn_0.3_O_4_ material, we have shown that we can introduce several dopants to propose a multi-level anti-counterfeiting compound in the visible and NIR ranges. (i) First, this work benefits from stable NIR-II photoluminescence from Er^3+^ and NIR-II persistent luminescence from Ni^2+^, in addition to NIR-I persistent emission when doping with Cr^3+^ (Figure 4 and Figure 6). (ii) Second, for further certification requirements, information and images could be read in both the visible and NIR ranges. To further address the advantages of anti-counterfeiting by this type of novel NP, patterns based on ZGSO:Ni^2+^, Er^3+^, Cr^3+^ NPs were evaluated by both Si-based and InGaAs-based cameras. A multi-level anti-counterfeiting system could then be obtained, as follows:

The changes in spectra obtained by optical spectroscopy (shown in Figure 6) are also observed in the images under both Si-based and InGaAs-based cameras. For example, the “MPOE” pattern was hand painted using ZGSO:Ni^2+^, Er^3+^, Cr^3+^ NPs. Figure 8 shows (a) an NIR-I persistent luminescence image from an “MPOE” pattern detected with a Si-based visible camera and (b) an NIR-II persistent luminescence image from an “MPOE” pattern detected with an InGaAs-based NIR camera after UV excitation removal. Appendix A shows (a) the “MPOE” pattern template for hand painting, using ZGSO-3:Ni^2+^, Er^3+^, Cr^3+^ NPs and (b) an NIR-II PL image from the “MPOE” pattern under UV excitation, detected with an InGaAs-based NIR camera with minimum contrast to avoid saturation. In comparison with the NIR-II PL image, the persistent luminescence signal takes advantage of avoidance of an autofluorescence background and provision of an enhanced anti-counterfeiting effect.

## 4. Conclusions

We successfully prepared several transition-metal (Ni^2+^ and/or Cr^3+^) cations and rare-earth (Er^3+^) co-doped ZGSO spinel nanocrystals with controllable photoluminescence wavelengths by tuning the concentration of Sn^4+^ and intensity emission and persistent luminescence for chromium and nickel transition-metal cations. Of these, multi-emission of PL in the NIR range benefits from the Er^3+^ ions working as emitters in nanoparticles of Zn_1.3_Ga_1.4_Sn_0.3_O_4_ co-doped with Ni^2+^, Er^3+^, and Cr^3+^ cations. Furthermore, Zn_1.3_Ga_1.4_Sn_0.3_O_4_ NPs exhibit persistent luminescence at 700 nm in the NIR-I range and 1300 nm NIR-II range, and their decay can last for more than 5 min after excitation removal.

A multi-level anti-counterfeiting pattern was designed by recording PL and persistent luminescence in the red, deep-red, and SWIR ranges. To further improve security, alternative excitation sources could be envisioned in the future due to the very wide excitation ranges of persistent luminescence materials [48,49,50,51,52].

## Figures and Tables

**Figure 1 materials-16-01132-f001:**
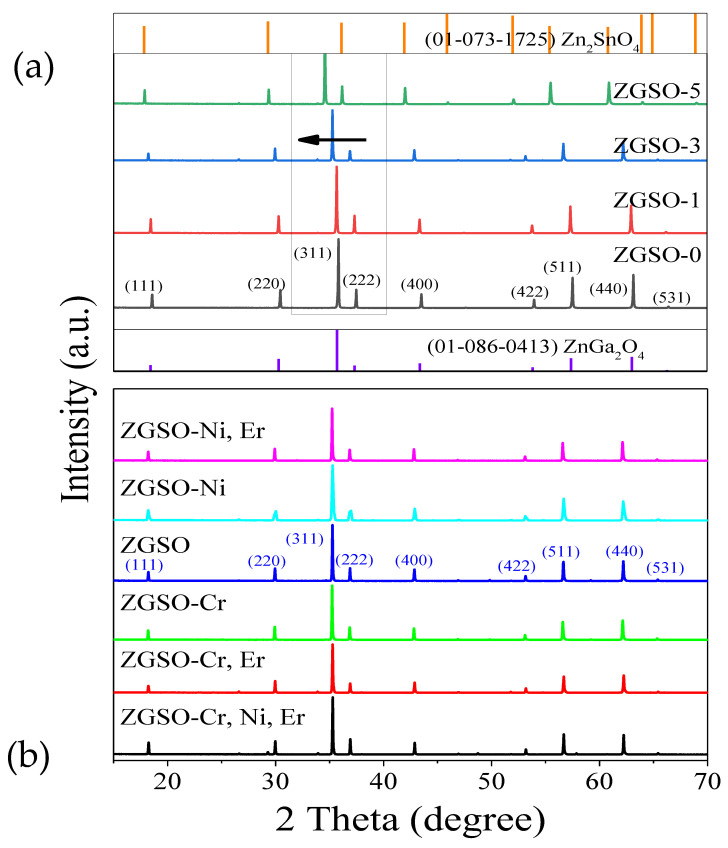
X-ray diffraction spectra of (**a**) ZGSO 0, 0.1, 0.3, and 0.5 concentration ratios of Sn/Ga matched to samples ZGSO-0, ZGSO-1, ZGSO-3, ZGSO-5, respectively. (**b**) Transition-metal (Ni^2+^ and/or Cr^3+^) and rare-earth (Er^3+^) ion-doped ZGSO-3.

**Figure 2 materials-16-01132-f002:**
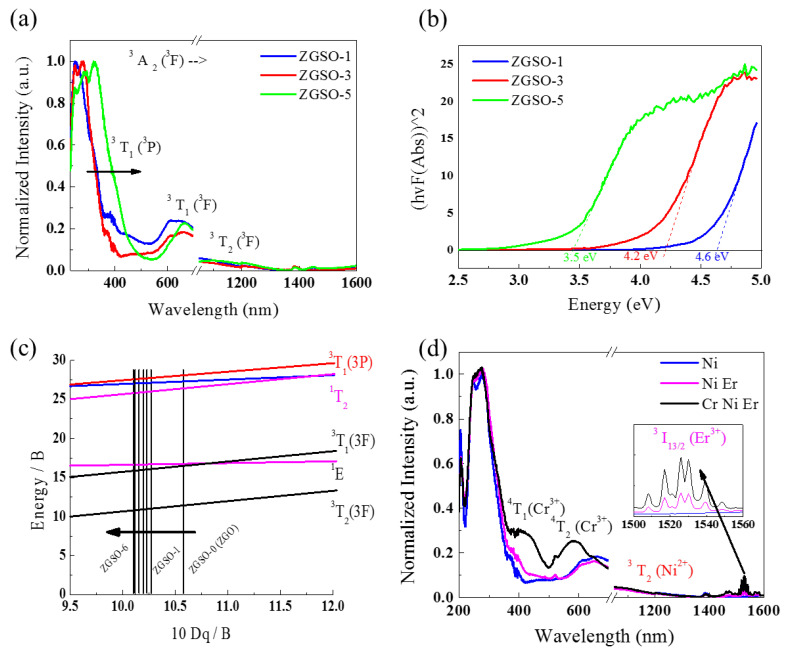
(**a**) Normalized absorption spectra of samples Ni^2+^-doped ZGSO-1 (Zn_1.1_Ga_1.79_Ni_0.01_Sn_0.1_O_3.995_), ZGSO-3 (Zn_1.3_Ga_1.39_Ni_0.01_Sn_0.3_O_3.995_), and ZGSO-5 (Zn_1.5_Ga_0.09_Ni_0.01_Sn_0.5_O_3.995_). (**b**) Bandgap calculation of samples Ni^2+^-doped ZGSO-1, ZGSO-3, and ZGSO-5. (**c**) Part of the Tanabe–Sugano diagram of Ni^2+^ with d^8^ electron configuration in the complex spinel samples of ZGSO-0~ZGSO-6 with different Sn concentrations (see Appendix A). (**d**) Transition-metal (Ni^2+^ and/or Cr^3+^) and rare-earth (Er ^3+^) doped ZGSO-3. The illustration is a partial enlargement of Er^3+^ absorption.

**Figure 3 materials-16-01132-f003:**
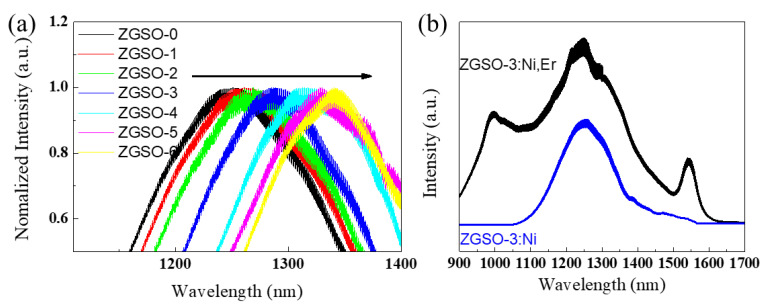
(**a**) PL spectra of 0.5% Ni^2+^-doped ZGSO-0 to ZGSO-6 samples. (**b**) PL of ZGSO-3:Ni^2+^ (blue) and ZGSO-3:Ni^2+^ Er^3+^ (black). All samples were excited using a 365 nm UV lamp.

**Figure 4 materials-16-01132-f004:**
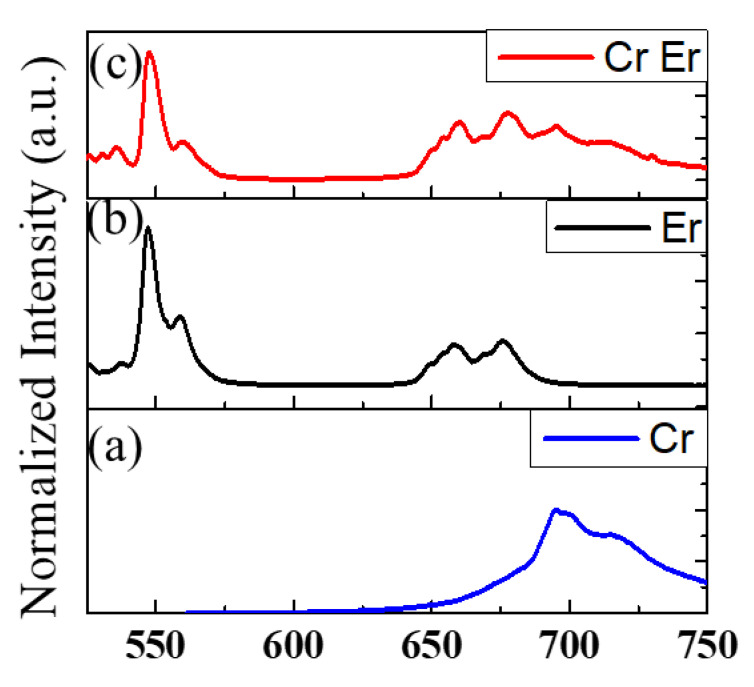
PL spectra of (**a**) ZGSO-3:Cr, (**b**) ZGSO-3:Er, (**c**) ZGSO-3:Cr, Er. All PL measurements are obtained from the samples under 365 nm UV excitation.

**Figure 5 materials-16-01132-f005:**
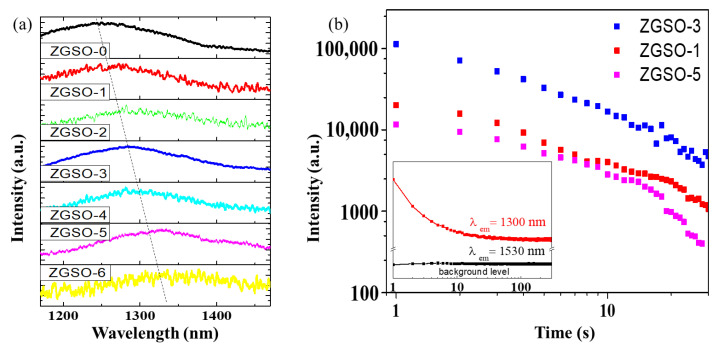
(**a**) Persistent luminescence spectra of the ZGSO-0–ZGSO-6 samples (1 min after ceasing UV excitation). (**b**) Corresponding decay curves of the ZGSO-1, ZGSO-3, ZGSO-5 samples after removing excitation. The inset presents the time dependence of the NIR persistent luminescence intensity of emission wavelength at 1300 nm and 1530 nm for Er^3+^, Ni^2+^ co-doped ZGSO-3. The emission at 1530 nm is at around the background level. All samples were excited using a 365 nm UV lamp for 5 min.

**Figure 6 materials-16-01132-f006:**
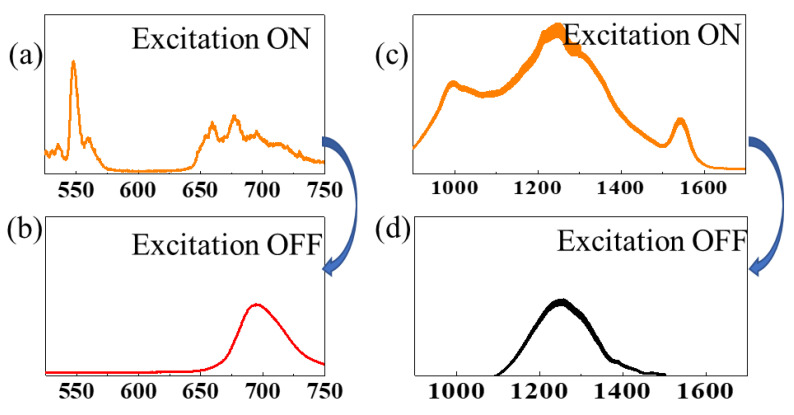
(**a**) Spectrum of PL (excitation ON) and (**b**) persistent luminescence (excitation OFF) detected with Si-based detector; (**c**) spectrum of PL (excitation ON) and (**d**) persistent luminescence (excitation OFF) detected with an InGaAs detector. All the PL or persistent luminescence spectra were from ZGSO-3:Ni, Cr, Er NPs under excitation or after excitation by the 365 nm UV lamp for 5 min, respectively.

**Figure 7 materials-16-01132-f007:**
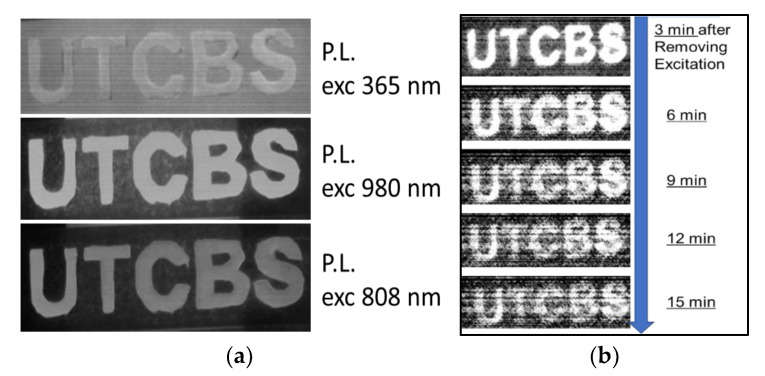
(**a**) PL of the “UTCBS” pattern noted by ZGSO-3:Ni^2+^, Er^3+^ NPs under excitation at 365 nm, 980 nm, and 808 nm, respectively. (**b**) Persistent luminescence images of the “UTCBS” pattern noted by ZGSO-3:Ni^2+^, Er^3+^ NPs after removal of 365 nm excitation for 5 min. All images were collected with an InGaAs camera.

**Figure 8 materials-16-01132-f008:**
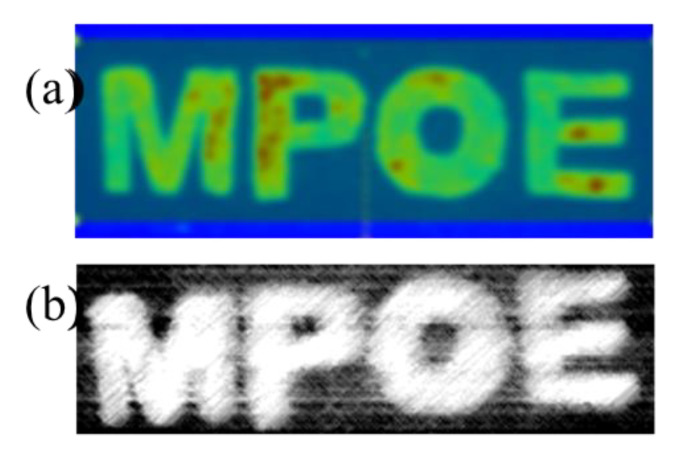
(**a**) Persistent luminescence image from the “MPOE” pattern, hand painted using ZGSO-3: Ni^2+^, Er^3+^, Cr^3+^ NPs detected with Si-based visible camera; (**b**) persistent luminescence image from the “MPOE” pattern, hand printed using ZGSO-3: Ni^2+^, Er^3+^, Cr^3+^ NPs, detected with InGaAs-based NIR camera.

**Table 1 materials-16-01132-t001:** Peak position of absorption and estimated crystal field parameters of Ni^2+^ in the spinel samples ZGSO.

No.	Sample/Material	^3^A_2_→^3^T_1_ [nm]	v _1_ [cm^−1^]	^3^A_2_→^3^T_1_ [nm]	v _2_ [cm^−1]^	^3^A_2_→^3^T_2_ [nm]	v _3_ [cm^−1^]	10 *D_q_* [cm^−1^]	B [cm^−1^]	10 *D_q_*/*B*
ZGSO-0	ZnGa_2_O_4_:0.5%Ni	370	27,027	622	16,077	1029	9718	9718	919	10.58
ZGSO-1	Zn_1.1_Ga_1.8_Sn_0.1_O_4_:0.5%Ni	371	26,954	625	16,000	1046	9560	9560	932	10.26
ZGSO-2	Zn_1.2_Ga_1.6_Sn_0.2_O_4_:0.5%Ni	372	26,882	628	15,924	1050	9524	9524	931	10.23
ZGSO-3	Zn_1.3_Ga_1,4_Sn_0.3_O_4_:0.5%Ni	374	26,738	632	15,823	1057	9461	9461	927	10.20
ZGSO-4	Zn_1.4_Ga_1.2_Sn_0.4_O_4_:0.5%Ni	376	26,596	636	15,723	1065	9390	9390	925	10.15
ZGSO-5	Zn_1.5_Ga_1_Sn_0.5_O_4_:0.5%Ni	379	26,385	641	15,601	1075	9302	9302	919	10.12
ZGSO-6	Zn_1.6_Ga_0.8_Sn_0.6_O_4_:0.5%Ni	382	26,178	646	15,480	1084	9225	9225	912	10.11

## Data Availability

The data presented in this study are available on request from the corresponding author.

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
