# Peer review of "ZGSO Spinel Nanoparticles with Dual Emission of NIR Persistent Luminescence for Anti-Counterfeiting Applications"

_materials, 2023, doi:10.3390/ma16031132_

Round 1
Reviewer 1 Report
Authors’ manuscript titled “Spinel Nanoparticles with Dual-Emission of NIR Persistent Luminescence for Anticounterfeiting Applications”, authors report the preparation of series of Zn1+xGa2-2xSnxO4 nanoparticles (ZGSO NPs) with persistent luminescence in the first and second near-infrared window. Doped ZGSO NPs exhibited intense photoluminescence in the range from red, NIR-I to NIR-II and even NIR-III under UV radiation, and showed persistent luminescence at 700 nm (NIR-I) and 1300 nm (NIR-II) after removing the excitation. The NPs were evaluated for multi-level anti-counterfeiting technology. This work is interesting.
1. “Persistent luminescence” abbreviated as “Persist. Lum.” is inappropriate in the manuscript. It can be suggested to abbreviate it as “PeL” for the convenience of readers.
2. In Page 2 Line 76-79, “According to the stoichiometric ratio of the compounds, For an example of ZGSO-3:Ni2+ (Zn1.3Ga1.4Sn0.3O4:0.5%Ni2+), 1.3 mmol of ZnO, 1.3 mmol of Ga2O3, 0.3 mmol of SnO2 and 0.1 mmol of NiO were weighed and ground homogenously in an agate mortar and the mixtures were introduced into alumina crucibles” is corresponding to “Ni2+ doped Zn1+xGa2-2xSnxO4 (ZGSO) matrix with various amounts of Sn (from x = 0 to x = 0.6).”(In Page 2 line 52-53), please explain how to calculate the stoichiometric ratio to obtain the “1.3 mmol of Ga2O3, 0.1 mmol of NiO”. How to get formula Zn1+xGa2-2xSnxO4(nanomaterials were designed as combinations of the normal spinel ZnGa2O4 and inverse spinel Zn2SnO4 with different Sn amounts, as represented by the formula Zn1+xGa2-2xSnxO4)?
3. In Page 3 line 107, The resolution is 1 nm for the Ce bands and 0.1 nm for the Gd and Cr bands? please check “the Ce bands”.
Author Response
Answers to reviewers 1 to 3

Reviewer 2 Report
The present manuscript reported a phosphor Zn1+xGa2-2xSnxO4:Ni2+ with long afterglow in NIR region, in which the afterglow was tuned through doping different Sn components. Furthermore, the Cr3+ and Er3+ were also introduced to obtain different wavelength emission, and these phosphors were also used in anti-counterfeiting field. However, there some important questions need to be answered.
1. In the host, which sites will be occupied by the dopants Ni2+, Cr3+ and Er3+? The corresponding experiments, explanation or theoretical calculations were not given. Please give enough data to support.
2. In the experimental, the contents of dopants Ni2+, Cr3+ and Er3+ must affect the emission intensity or afterglow intensity, but authors did not give related investigation process or results.
3. The introduction of Cr3+ and Er3+ will or not affect the afterglow of Ni2+? Please give related experimental results.
4. Please give the PLE spectra of the afterglow, furthermore, the mechanism of the afterglow should also been investigated by comparison the TL spectra of Ni2+, Cr3+ and Er3+ solely, codoped or tri-doped samples.
5. Under the excitation of 980 nm laser, no upconversition was occurred in Er3+ doped samples? More recent papers have been published in anti-counterfeiting field or related field are useful for the present paper, such as Materials Today Physics 2022,27:100830, Ceramics International 2021,47(11):15067-15072.
Author Response
answers to all reviewers

Reviewer 3 Report
The article undoubtedly contains some new results that may be recommended for publication, but only after improvement and concretization of some incomprehensible points.
1. The title is unfortunate and misleading. The word spinel refers to a magnesium-aluminum compound, therefore it is proposed (if we are talking about the structure) to specify the title. For example as “ZnGa2O4 Spinel Nanoparticles with Dual-Emission of NIR Persistent Luminescence for Anticounterfeiting Applications”
2. ZnGa2O4 doped with luminescent ions is the main object of this paper. Therefore, more attention in the Introduction should be paid to this compound, its properties and its applications. See, for example, a few recent reports:
Luchechko, A., Zhydachevskyy, Y., Ubizskii, S. et al. Afterglow, TL and OSL properties of Mn2+-doped ZnGa2O4 phosphor. Sci Rep 9, 9544 (2019). https://doi.org/10.1038/s41598-019-45869-7
Zhang, X., Zhang, J., & Zhu, Q. (2022). Doping upconversion ion pair of Yb3+/Er3+ in ZnGa2O4: Cr3+ for multimode luminescence and advanced anti-counterfeiting. Optical Materials, 125, 112100.
and references therein.
3. Can one still justify drawing a straight line to determine Eg (Fig.2). The definition of the band gap Eg here is quite ambiguous. See the latest article by the editors of Optical Material: M.G. Brik, A.M. Srivastava et al, A few common misconceptions in the interpretation of experimental spectroscopic data. Optical Materials 127 (2022) 112276 https://doi.org/10.1016/j.optmat.2022.112276.
4. how free are these nanocrystals from point defects and what role would they play if their concentration were sufficient to influence these luminescent processes
In principle, this is a rather interesting topic, which, of course, needs to be developed and promoted, the results obtained are interesting and can be recommended for publication after detailed consideration and disclosure of the above-mentioned ambiguities.
Author Response
answers

Round 2
Reviewer 2 Report
All raised questions have been anwered, it could be accepted.
Reviewer 3 Report
After quite successful revision, this manuscript can be accepted